# School-aged *Schistosoma mansoni* infection levels after long-term programmatic control show failure to meet control programme targets and evidence of a persistent hotspot: evaluation of the FibroScHot trial baseline data

Fred Besigye[1], Candia Rowel[1], Moses Adriko[1], Fredrick J. Muyodi[2],
John Joseph Kisakye[2], Rosemary Nalwanga[2], Birgitte J. Vennervald[3], Fred Nuwaha[4],
Edridah M. Tukahebwa[1], Shona Wilson [5]*

1 Vector Control Division, Ministry of Health, Kampala, Uganda, 2 Department of Zoology, Entomology and Fisheries Sciences, College of Natural Sciences, Makerere University, Kampala, Uganda, 3 Section for Parasitology and Aquatic Pathobiology, University of Copenhagen, Denmark, 4 School of Public Health, College of Health Sciences, Makerere University, New Mulago Hill, Mulago, Kampala, Uganda, 5 Department of Pathology, University of Cambridge, Cambridge, United Kingdom

* sw320@cam.ac.uk

## Abstract

### Background

Treatment guidelines for schistosomiasis recommend increasing frequency of preventative chemotherapy (PC) administration of praziquantel to twice per annum in persistent hotspots of transmission, in combination with integrated control strategies. FibroScHot was an individual randomised superiority trial designed to examine twice per annum and four times per annum treatment frequency. It was conducted in two primary schools, Buhirigi and Kaiso, in Hoima District Uganda – a designated *Schistosoma mansoni* high transmission area in which PC is targeted at children and adults. The baseline parasitology data was assessed against international control programme thresholds of success and the criteria for persistent hotspots. Further, the study also assessed the potential for integrated control strategies within the surrounding communities.

### Methodology/principal findings

The prevalence of infection, heavy infection and the infection intensity were derived for 700 participants from Kato-Katz examination of one stool sample. Neither school met the threshold of morbidity control (<5% with heavy infection). A strong school effect was observed in models of prevalence and prevalence of heavy infection, with these being greater in Kaiso. By prevalence, Kaiso was a high transmission area and Buhirigi a moderate transmission area. Kaiso but not Buhirigi met the definition of a

**Data availability statement:** All data that support the findings of this study are available from UK DataService ReShare: Besigye, Fred and Adriko, Moses and Vennervald, Birgitte J and Tukahebwa, Edridah M and Bond, Simon and Wilson, Shona (2024). Impact of Increased Praziquantel Frequency on Childhood Fibrosis in Persistent Schistosomiasis Morbidity Hotspots (FibroScHot ) Baseline Parasitology and WASH Infrastructure: 2019 - 2021. [Data Collection]. Colchester, Essex: UK Data Service. 10.5255/UKDA-SN-857375

**Funding:** Data collection and authors FB, MA, BJV, FN, EMT and SW were funded by the FibroScHot project which is part of the EDCTP2 programme supported by the European Union (RIA2017NIM-1842-FibroScHot). www.edctp.org The funders had no role in the study design, data collection and analysis, decision to publish, or preparation of the manuscript.

**Competing interests:** The authors have declared that no competing interests exist.

persistent hotspot. Persistent hotspot classification did not change when intensity of infection was used. Intermediate snail hosts were collected at both Kaiso landing site and from the River Hoimo in Buhirigi, though in smaller numbers in the latter. Questionnaire data indicates that reliance on water collection from transmission sites and open defecation occurs more frequently in Kaiso than in Buhirigi.

## Conclusions

The criteria for persistent hotspots were met in the high transmission but not the moderate transmission community despite neither community meeting the threshold of morbidity control. This disconnect indicates that endemic communities exist in which control has not been achieved but increased frequency of treatment is currently not recommended. FibroScHot will be able to inform on whether widening the current recommendation of increased treatment frequency to these communities will achieve improved control. Evidence provided also indicates scope for the integrated control strategies of vector control and WASH improvements in both the participating communities.

### Authors summary

Schistosomiasis is a highly significant parasitic disease. Control programmes administering the drug praziquantel on an annual basis to school age children, and to adults in areas of high-risk, aimed primarily to prevent severe morbidity through significant reduction of the burden of infection (morbidity control). Despite success in many communities, it has been systematically shown that infection persists in others despite good treatment coverage rates by the control programmes. These communities are known as persistent hotspots. In response WHO recommends increasing treatment frequency to twice per annum; though caveats of limited evidence both in the definition used for hotspot detection, and in the likely success of the twice per annum strategy, particularly for *Schistosoma mansoni*, exist. The FibroScHot trial aimed to assess this twice per annum strategy but also a more intensive 4x per annum strategy. Crucial to interpretation of the trial results will be establishment of whether or not the trial was undertaken in persistent hotspots. Evidence presented here indicates that one trial site but not the second meets the current definition of a persistent hotspot, despite neither having met the threshold of morbidity control.

## Introduction

Schistosomiasis is a highly significant Neglected Tropical Disease (NTD), endemic in 72 countries. The NTD Roadmap for 2021 – 2030 aims to eliminate the disease

as a public health problem (EPHP; defined as <1% with heavy infections) in all endemic countries by 2030 [1]. Prior to this the commitment was to control schistosomiasis morbidity (defined as <5% with heavy infections) [2]. Morbidity control was to be achieved by implementing preventive chemotherapy (PC) through mass drug administration (MDA), targeted at 75–100% of school going children (SAC) and adults at risk, with the anti-helminthic praziquantel [3]. The latest Roadmap calls for integration with hygiene and sanitation health promotion, water and sanitation (WASH) infrastructure improvement and intermediate snail host control in order to meet the tougher aim; and treatment guidelines released in 2022 call for expanded treatment-based strategies in consort with these integrated strategies [4].

Operational hotspots in which infection levels fail to respond to PC due to implementation issues occur, but also biological hotspots in which parasite and host factors drive transmission despite good treatment coverage [5]. Control of infection in biological hotspots is essential if EPHP is to be achieved. The expanded guidelines recognise this with a specific recommendation to increase treatment frequency to twice per annum in *persistent* biological hotspots [4]. Crucial to targeting this increased treatment effectively is a clear strategy by which persistent hotspots can be identified. Persistent hotspots are difficult to detect from baseline data, a process not significantly improved for *S. mansoni* using machine learning and secondary environmental data [6], so field surveys are required. Calculating relative changes in prevalence rather than absolute prevalence data have come to prominence, largely through the Schistosomiasis Consortium for Operational Research and Evaluation (SCORE) consortium [7]. Initially, a failure to decrease in prevalence by ~35% after a lengthy 5-years of ≥75% MDA coverage was used to classify persistent hotspots [8], but analysis shows they can predicted by year 3 [9], from which the following 4-criteria definition has arisen: i) a baseline prevalence of ≥10%; ii) at least 2 rounds of PC conducted; iii) coverage at ≥75% of targeted population; iv) a reduction in prevalence of <1/3.

In Uganda, schistosomiasis is a common neglected tropical disease [10], present in 43 of 73 districts [11,12]. Intestinal schistosomiasis caused by *S. mansoni* is predominant, with 5.4 million individuals estimated to be infected and 13.9 million are at risk of acquiring the infections. Uganda was at the forefront of PC control programme implementation, starting in highly endemic lake shore communities in 2003 [13] and success on a macro country-wide scale in gaining morbidity control has been established [14]. However, the national infection prevalence remains high, estimated at 25.6% in 2018 [15] and studies in lakeshore communities pre- and post- control programme implementation suggest that they are highly likely to be persistent hotspots [16–19].

FibroScHot was a phase IV open label individual randomised superiority trial designed to determine whether 2x per annum, or a more intensive 4x per annum PC for SAC, can provide a treatment strategy for controlling schistosomiasis morbidity. It has been conducted in two schools situated in contrasting communities in Hoima District, western Uganda. Here we present the baseline parasitology findings, compare them with WHO definitions of transmission areas, control programme success and by utilising the findings of a 2016 monitoring exercise, determine whether the communities are classifiable as persistent hotspots. This information will be crucial in the interpretation of the FibroScHot trial but also informs on the applicability of the current definition of persistent hotspots. We also provide evidence for integrated control strategies for the two communities.

## Methods

### Ethics statement

The FibroScHot research programme received ethical approval from the Vector Control Division Research Ethics Committee (VCDREC110), the University of Cambridge Human Biology Research Ethics Committee (HBREC.2018.32) and the Uganda National Council for Science and Technology (UNCST HS2625). Clinical trial approval and certification was obtained from the Uganda National Drug Authority (CTC 0122). The trial was prospectively registered with the International Standard Randomized Controlled Trial Number (ISRCTN) Registry (ISRCTN 16994599; https://doi.org/10.1186/ISRCTN16994599). Written parental/guardian consent and participant assent was obtained for all participants.

## Study area

The trial was conducted in Kaiso and Buhirigi Primary Schools located in the Hoima District sub-counties of Kabaale and Kigorobya, respectively. Kaiso is on the lake shore, while Buhirigi is located above the escarpment approximately 30km from Kaiso as the crow flies (Fig 1). The schools were selected on four criteria: 1) representative of high infection intensity schools as determined from pre-MDA monitoring in October 2016; 2) stability of residence, assessed using place of birth data, coupled with teacher reported stability of the pupil population; 3) school enrolment numbers and 4) relatively close proximity to each other.

Prior to 2020 a total of 11 rounds of MDA targeting SAC and adults had been conducted in Hoima District. Coverage rates are recorded in the Expanded Special Project for Elimination of Neglected Tropical Diseases (ESPEN; https://espen.afro.who.int) from 2014 onwards and in all years >75% of the targeted SAC were treated, with the exception of 2014 when 74.6% were treated. This is corroborated by internal data held by the Uganda Ministry of Health. There is no treatment data recorded in ESPEN for 2016, but the internal Ministry of Health data indicates that MDA was conducted with 81.8% of targeted SAC treated. A twelfth round of MDA, recorded in ESPEN for 2019 only targeted SAC. This round was actually conducted in June 2020 due to delayed receipt of praziquantel in Uganda and the COVID-19 pandemic.

## Study design and participants

The trial aimed for a sample number of 600 study participants entering the treatment phase. To be eligible children were aged 6–14 years of age at enrolment, had been born or resident within the community for a minimum of 2 years and

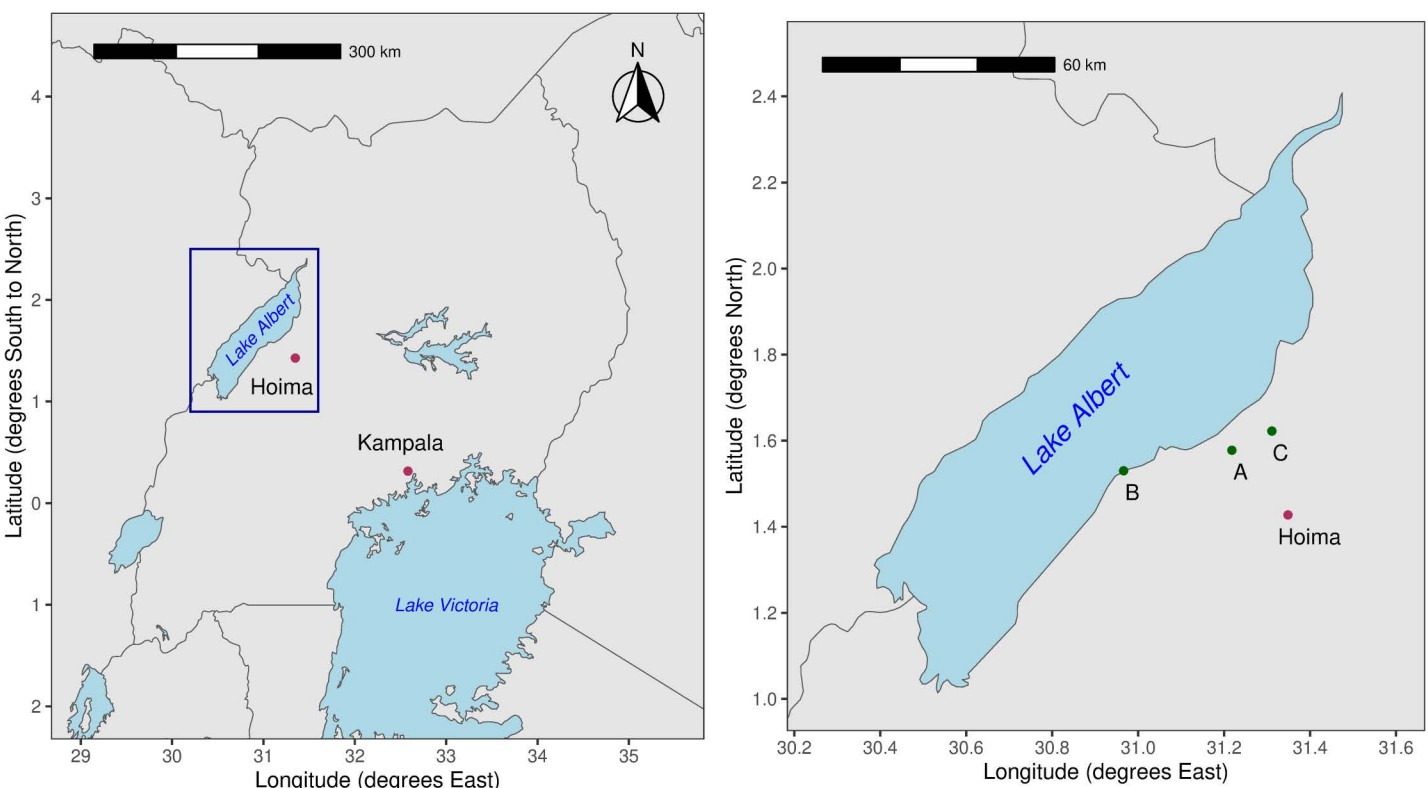

**Fig 1. Study Site.** Location of the two schools participating in the study in relation to Kampala, Hoima Town and Lake Albert. A: Buhirigi Primary, B: Kaiso Primary and C: Kigorobya Town where the FibroScHot trial facility for document and drug storage was situated. Base maps were generated in R software using the open source rnaturalearth data package (https://www.naturalearthdata.com), using the country shape files for Uganda, the Democratic Republic of the Congo, Kenya, Rwanda, South Sudan, and the United Republic of Tanzania and the Lakes shape file download.

voluntary parent/guardian consent and participant assent obtained. Children who had a history of facial oedema after treatment with praziquantel or known neurocysticercosis or intraocular cysticercosis were excluded. Enrolment into the trial was conducted on an all-comers basis, with initial recruitment from the P2 and P3 classes, with later expansion into other classes. Enrolment was split between 2020 and 2021 due to a pause caused by the COVID-19 pandemic. To ensure that sample numbers were maintained, catch up enrolment was conducted in 2021, resulting in baseline parasitology data for 700 participants. All parasitology sampling for Kaiso Primary was conducted in 2021, parasitology sampling for Buhirigi Primary was split between 2020 and 2021. The round of MDA conducted in June 2020 occurred between the split baseline visits. As it would have been unethical to withhold participants from this treatment, the 2020 examinations occurred 18-months and the 2021 examinations 9-months after the previous round of MDA.

*Schistosoma mansoni* infection data for the schools in 2016 were available from an internal monitoring exercise conducted by the Uganda Ministry of Health under their remit to monitor infection and morbidity in response to control programme implementation.

## Sample collection and processing

For trial purposes multiple stool samples were collected and processed using the Kato Katz (KK) method [20,21], with two slides prepared from each sample where possible. Here, in order to compare baseline results with WHO recommendations for control programme monitoring activities, results were restricted to the mean of the egg counts obtained from the two slides prepared from the first stool sample. Raw egg counts were converted into eggs per gramme of faeces (epg) prior to analysis.

## Status of WASH infrastructure

The 2018 Core questions on the household drinking-water and sanitation from the joint monitoring programme (JMP) for water supply and sanitation questionnaire developed by WHO and UNICEF [22] was adopted and translated into Alur (the main local language). It was implemented in November 2019. Parents/guardians of children attending the P2 class in each school were invited to participate and were identified by asking a parent/guardian of every 4th child on the register. Fifty parents/guardians from Kaiso and 48 parents/guardians from Buhirigi participated. They were provided with a study code that was not linked to their child.

## Malacology

Three 30-minute malacology sweeps at a depth of 0.5-1m were conducted at four sampling sites at the Kaiso landing site on Lake Albert and three on River Hoimo in Buhirigi community. Collected snails were sorted to the genus *Biomphalaria* and identified to species [23] prior to counting. Individual snails were then exposed to indirect natural light for a minimum of 30 minutes and up to a maximum of 2 hours and screened for shedding of human schistosome cercariae. The cercariae present were identified by general morphological/anatomical appearance using standard taxonomic keys [24] and the species of each shedding *Biomphalaria* snail recorded.

## Data analysis

The following demographic data: sex; age divided into 6–8, 9–11 and 12–14 years of age; and length of residence divided into 2–4 years and > 4-years; were compared between school attended, and for Buhirigi Primary only, the year of baseline examination, using Chi-square testing. Age as a continuous variable was compared by t-test.

Egg count data was analysed as a linear variable, but also collapsed into two binomial variables, one representing yes/no infected (prevalence of infection) and one representing non-heavy and heavy infection according to the WHO definition of ≥400 epg of faeces being a heavy infection (prevalence of heavy infection). Prevalence data was compared to WHO definitions of transmission for control programme implementation design: high transmission (≥50% amongst SAC);

moderate transmission (≥10% but <50% amongst SAC) and the relative reduction in prevalence and infection intensity from the 2016 monitoring was calculated in line with the WHO treatment guidelines [4]. Prevalence of heavy infection was compared directly with the WHO programmatic thresholds of <5% - morbidity control, and <1% - EPHP [1]. Demographic predictors of prevalence and prevalence of heavy infection were analysed by logistic regression. To examine predictors of intensity of *S. mansoni* infection amongst those with a detectable infection, linear regression models were built with the log transformed egg count data as the dependent variable.

Categorical data from the WASH questionnaire with less than five responses were collapsed into "alternative" for analysis. Chi-squared analysis was conducted to determine significant differences between the two communities in recorded replies. The number of households sharing sanitation facilities was analysed using a Man-Whitney test and estimated time spent collecting water was analysed by t-test. Descriptive results only are presented for the malacology data. In all analysis an alpha of 5% ($p < 0.05$) was considered significant.

## Results

### Demography of participants

The demographics by school and by year are shown in Table 1. For Buhirigi, those recruited in 2021 were significantly younger (t = 2.450, p = 0.016) and were significantly more likely to have lived in the community for 2–4 years rather than >4-years than those recruited in 2020 ($x^2$ = 298.23, p < 0.001). Participants from Kaiso Primary were younger than participants from Buhirigi (t = 2.568, p = 0.010), but the actual mean age difference is negligible at less than 5-months of age.

### *Schistosoma mansoni* infection levels

The prevalence of infection, prevalence of heavy infection and the infection intensities for the 2016 monitoring exercise and the trial baseline are shown in Table 1.

**Prevalence of *S. mansoni* infection.** Given the prevalence at trial baseline, Kaiso can be characterised as a high prevalence setting; whilst regardless of year of examination, Buhirigi Primary meets the definition of a moderate prevalence setting. For Buhirigi, overall, there was a 42.6% reduction in prevalence since the 2016 monitoring. For those examined in 2020 the reduction was 37.4% and for those examined in 2021 the reduction was 60.8%. For Kaiso, there was a 31% increase in prevalence since the 2016 monitoring. Kaiso can therefore be considered a persistent hotspot under WHO criteria while this is not the case for Buhirigi. In logistic regression analysis conducted to determine the demographic predictors of having a detectable *S. mansoni* infection, school and age group were the only two significant predictors, with prevalence increasing with age group. The school effect is of considerable magnitude. Year of examination was of borderline significance (Table 2).

**Intensity of *S. mansoni* infection.** Overall, in Buhirigi, a relative reduction in infection intensity of 79.5% since 2016 was recorded; among participants enrolled in 2020 the relative reduction was 74.8% and for participants enrolled in 2021 it was 91.5%. In Kaiso a relative increase of 56.3% in infection intensity since 2016 was recorded. Regardless of school and year of examination, there was a significant proportion of participants with detectable *S. mansoni* infections whose infection intensity was classifiable as moderate or heavy (Fig 2). In a linear regression model no demographic factors were found to be predictive of the intensity of infection amongst those infected (results not shown).

**Prevalence of heavy *S. mansoni* infection.** The threshold of morbidity control (<5% with a heavy infection) was not met in either school, regardless of year of examination (Table 1). Logistical regression analysis shows that being in the oldest age group (12–14years) and attending Kaiso Primary were significantly associated with having a heavy infection (Table 3).

### Sources and treatment of drinking water

The sources of water differed significantly between the two communities ($x^2$ = 41.626, p < 0.001). Amongst respondents from Buhirigi community the most common source of water was boreholes (Table 4), whereas in the Kaiso community the

**Table 1. Baseline demographic characteristics and parasitology summary statistics by school and year of examination.**

**2016 Monitoring Exercise**

|  |  | Kaiso | Buhirigi |  |  |
|---|---|---|---|---|---|
|  |  | (n = 51) | (n = 49) |  |  |
| *S. mansoni* Prevalence | Negative (n (%)) | 20 (39.2) | 16 (32.7) |  |  |
|  | Positive (n (%)) | 31 (60.8) | 33 (67.3) |  |  |
| Prevalence of *S. mansoni* infection intensity groupings[1] | Light (n, (%)) | 2 (0.0) | 14 (28.6) |  |  |
|  | Moderate (n (%)) | 12 (23.5) | 13 (26.5) |  |  |
|  | Heavy (n, (%)) | 17 (33.3) | 6 (12.2) |  |  |
| *S. mansoni* infection intensity (geometric mean (±95% C.I.; maximum)). |  | 40.6 epg (16.3, 99.0; 3240) | 26.1 epg (12.0, 55.7; 4020) |  |  |

**Baseline examinations**

|  |  | Kaiso (n = 286) | Buhirigi combined (n = 414) | Buhirigi 2020 (n = 323) | Buhirigi 2021 (n = 91) |
|---|---|---|---|---|---|
| Sex | Female (n, (%)) | 117 (40.9) | 198 (47.8) | 149 (46.1) | 49 (53.9) |
|  | Male (n, (%)) | 169 (59.1) | 216 (52.2) | 174 (53.89) | 42 (46.2) |
| Age | Mean (± 2 S.E.) | 9.7 (0.25)*[2] | 10.1 (0.22) | 10.3 (0.23) | 9.57 (0.54)*[3] |
| Age Group (years) | 6-7 (n, (%)) | 87 (30.4) | 112 (27.1) | 77 (23.8) | 35 (38.5) |
|  | 8-10 (n, (%)) | 128 (44.8) | 162 (39.1) | 132 (40.9) | 30 (33.0) |
|  | 11-14 (n, (%)) | 71 (24.8)*[2] | 140 (33.8) | 114 (35.3) | 26 (28.6)*[3] |
| Residency (years) | 2-4 (n, (%)) | 42 (14.7) | 73 (17.6) | 1 (0.3) | 72 (79.1) |
|  | >4 (n (%)) | 244 (85.3) | 341 (82.4) | 322 (99.7) | 19 (20.9)***[3] |
| Prevalence of *S. mansoni* infections | Negative (n, (%)) | 58 (20.3) | 254 (61.4) | 187 (57.9) | 67 (73.6) |
|  | Positive (n, (%)) | 228 (79.7) | 160 (38.6) | 136 (42.1) | 24 (26.4) |
| Prevalence of *S. mansoni* infection intensity groupings | Light (n, (%)) | 81 (28.3) | 78 (18.8) | 64 (19.8) | 14 (15.4) |
|  | Moderate (n (%)) | 74 (25.9) | 39 (9.4) | 34 (10.5) | 5 (5.5) |
|  | Heavy (n, (%)) | 73 (25.5) | 43 (10.4) | 38 (11.8) | 5 (5.5) |
| *S. mansoni* infection intensity (geometric mean (±95% C.I.; maximum)). |  | 63.5 epg (47.7, 84.4; 3588) | 5.3 epg (4.0, 7.1; 3228) | 6.6 epg (4.8, 9.1; 3228) | 2.2 epg (1.1, 4.0; 1116) |

* p < 0.05, ***p < 0.001. [1]Infection intensity classified in accordance to WHO guidelines: light (1–99epg), moderate (100–399epg) and heavy (≥400epg). [2]Comparison with Buhirigi combined, [3]comparison with Buhirigi 2020. The 2016 monitoring was conducted in October immediately prior to MDA implementation.

most common source of water was "other". Observation and anthropological investigation reported elsewhere indicate that this "other" is the lake [25]. The participants from Buhirigi community reported more time spent in collecting water from the source (mean = 32.10 minutes, 95%CI: 27.87, 36.33) compared to those from Kaiso community (mean = 24.54 minutes, 95% CI: 19.42, 29.62; t = 2.835, p = 0.006). However there was no significant difference in who collected water, with 93.6% of respondents in Buhirigi and 92% of respondents in Kaiso reporting that water was collected by adult females ($\chi^2 = 0.095$, *p* = 0.758). The remaining individuals all reported that water was collected by females under the age of 15. Replies regarding if and how this water was treated indicated that safe water practices were more common in Buhirigi (S1 Fig).

## Sanitation use in Buhirigi and Kaiso communities

Open defecation was reported by 5 (10%) of those in Kaiso community, but by none in Buhirigi community. The sharing of latrines with other households was more common in Buhirigi (41 (68.3%)) than in Kaiso (19 (31.7%); $\chi^2 = 20.658$,

**Table 2. Logistic Regression analysis of *S. mansoni* infection detectable by parasitological method.**

| Variable | Odds Ratio (95% CI) | p-value |
|---|---|---|
| **Age 9–11 years** | 1.740 (1.154, 2.635) | 0.008 |
| **Age 12–14 years** | 2.829 (1.812, 4.459) | <0.001 |
| **Sex** | 0.838 (0.597, 1.173) | 0.304 |
| **School at enrolment** | 12.699 (6.388, 26.739) | <0.001 |
| **Residency** | 0.909 (0.455, 1.727) | 0.779 |
| **Year of examination** | 0.491 (0.226, 1.013) | 0.062 |

Reference groups are: Aged 6–8 years, Females, Buhirigi Primary, 2–4 years residency and examined in 2020.

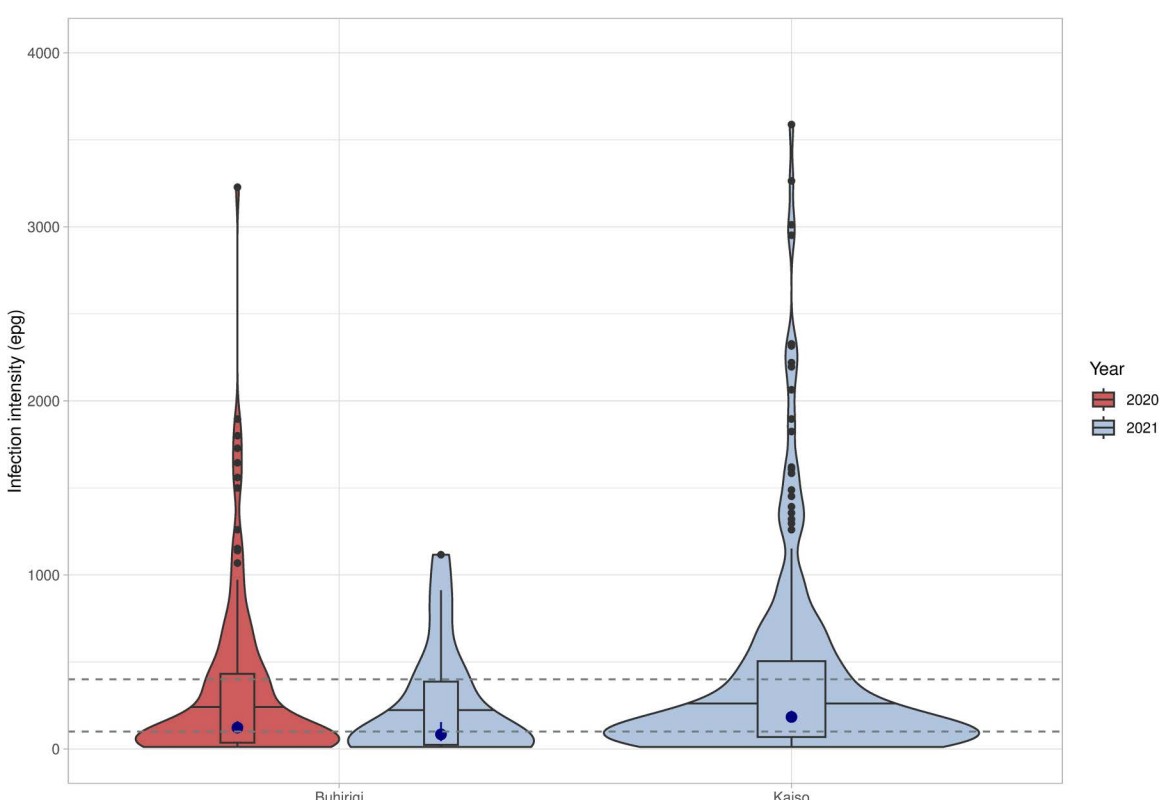

**Fig 2. *Schistosoma mansoni* infection intensities.** The distribution of the infection intensities (violin plots) is shown for participants who had an infection detectable by parasitological method within the two schools, and for Buhirigi by year of examination. Solid lines across the violin plots show the median infection intensity, boxplots display the interquartile ranges and grey circles the outliers. A closed blue circle indicating the geometric mean and 95% confidence intervals are shown within the boxplots. The two grey dotted lines indicate the thresholds for moderate and heavy infections according to WHO definitions.

$p < 0.001$). However, when latrines were shared, the number of households reported to be using each latrine was less in Buhirigi community (Range 1–15 HH, median = 6) than in Kaiso community (Range = 2–20 HH, median = 7; U = 267.5, $p = 0.009$). Forty-one (85.6%) respondents from Buhirigi community and 40 (80%) respondents from Kaiso community had infants below three years of age ($\chi^2 = 0.501$, $p = 0.479$). Participants in the two communities reported different ways of disposing of their infants' faecal waste. In Buhirigi only one reported disposing of their infant's faeces by burying them in

**Table 3. Logistic Regression analysis of heavy *S. mansoni* infection detectable by parasitological method.**

| Variable | Odds Ratio (95% CI) | p-value |
|---|---|---|
| **Age 9–11 years** | 1.234 (0.730, 2.123) | 0.439 |
| **Age 12–14 years** | 2.010 (1.165, 3.531) | 0.013 |
| **Sex** | 1.208 (0.797, 1.842) | 0.378 |
| **School at enrolment** | 6.479 (2.411, 20.906) | <0.001 |
| **Residency** | 0.873 (0.433, 1.834) | 0.712 |
| **Year of examination** | 0.418 (0.122 1.205) | 0.129 |

Reference groups are: Aged 6–8 years, Females, Buhirigi Primary, 2–4 years residency and examined in 2020.

**Table 4. Reported water sources in Buhirigi and Kaiso Communities.**

| Water Sources | Buhirigi | Kaiso | Total |
|---|---|---|---|
| Borehole | 23 (47.9%) | 3 (6%) | 26 (26.5%) |
| Reported as "Other" | 2 (4.2%) | 24 (48%) | 26 (26.5%) |
| Standpipe | 5 (10.4%) | 4 (8%) | 9 (9.2%) |
| Surface water | 9 (18.8%) | 9 (18%) | 18 (18.4%) |
| Unprotected spring | 0 (0.0%) | 8 (16%) | 8 (8.2%) |
| Alternative source* | 9 (18.8%) | 2 (4%) | 11 (11.2%) |
| **Total** | **48 (100%)** | **50 (100%)** | **98 (100%)** |

*Sources reported by < 5 individuals in both communities were grouped as an "alternative source".

ground, with 92.5% reporting rinsing them into toilet/latrine. By contrast, in Kaiso 19 (47.5%) respondents reported burying the faeces ($\chi^2 = 25.806$, $p < 0.001$; Fig 3).

## Intermediate host population numbers and patent infections

Permissive intermediate host snail species of the genus *Biomphalaria* were collected at all seven sites sampled. In total 1566 snails of the *Biomphalaria* genus were collected; their distribution by community and site is shown in Table 5. Amongst the Kaiso collections, seven snails were found to shed human infective cercariae. Three were *B. sudanica* collected from site C and site D; three were *B. pfeifferi,* all collected from site C and the seventh was of the species *B. stanleyi* and was collected from site D. No *Biomphalaria* spp snails collected from the River Hoimo in Buhirigi were found to shed human infective *Schistosoma spp.* cercariae.

## Discussion

The guideline to increase frequency of praziquantel treatment of schistosomiasis to twice per annum in hotspots has been provided on a "conditional" basis reflecting uncertainty in persistent hotspot classification [4]. Here, we assessed the baseline parasitology data from the FibroScHot trial to establish whether the participating communities met the current criteria for persistent hotspots. The 4-criteria definition was clearly met for Kaiso Primary. In fact, prevalence increased since the 2016 monitoring and was comparable with an ESPEN record from 2011 (79.6%), indicating little to no impact of control on prevalence of infection in the decade from 2011 – 2020. This is despite examinations in Kaiso taking place only 9-months after the previous PC treatment. The lower prevalence and infection intensities in the Kaiso 2016 data may reflect a risk that the evaluation unit of 50 children per school fails to capture true infection levels. The limited evidence on the optimal survey sample size for hotspot determination is recognised by WHO [26]. For Buhirigi Primary, a > 1/3 relative reduction in

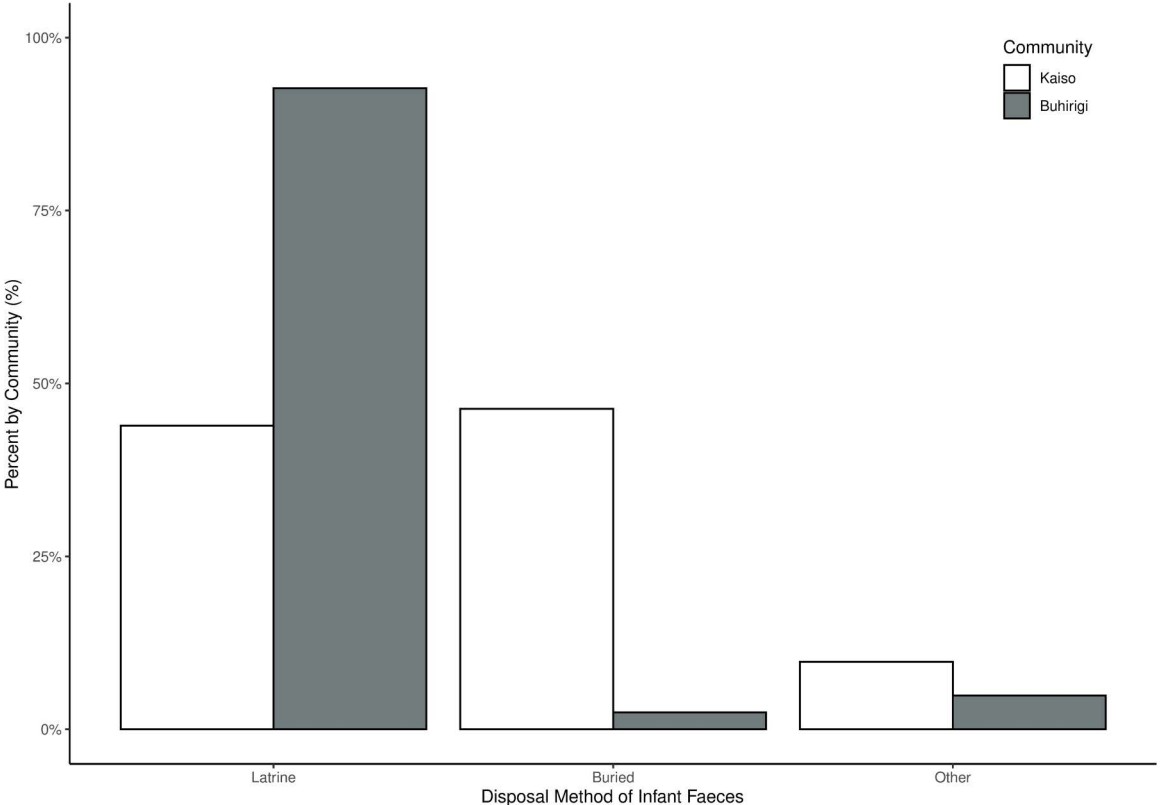

**Fig 3. Ways by which infants' faecal material were disposed of in Buhirigi and Kaiso communities.** Shown are the percentage of individuals reporting each way of faecal disposal from amongst those who answered "yes" to having children <3 years. Only answers with >5% of respondents in either community are displayed, with other responses grouped as "other".

**Table 5. Abundance of intermediate hosts at collection sites on the River Hoimo in Buhirigi Community and Lake Albert in Kaiso Community.**

| Village | Site | *B. sudanica* | *B. stanleyi* | *B. pfeifferi* | Total |
|---------|------|---------------|---------------|----------------|-------|
| **Kaiso** | A | 0 | 4 | 0 | 4 |
| | B | 12 | 2 | 17 | 31 |
| | C | 535 | 42 | 705 | 1282 |
| | D | 12 | 67 | 35 | 114 |
| **Sub-Total** | | **559** | **115** | **757** | **1431** |
| **Buhirigi** | A | 0 | 0 | 114 | 114 |
| | B | 0 | 0 | 2 | 2 |
| | C | 0 | 0 | 19 | 19 |
| **Sub-Total** | | **0** | **0** | **135** | **135** |
| **TOTAL** | | **559** | **115** | **892** | **1566** |

prevalence between the 2016 monitoring and the 2020/21 baseline was observed. It cannot be ascertained whether this reduction would have been achieved after 2-years of treatment as set-out as minimum in the WHO definition, as three treatment rounds had been conducted (including the 2016 round not recorded in ESPEN). Due to the complexity of identifying persistent hotspots, consideration of relative differences in intensity in combination with those in prevalence has

been called for [5]. For both Kaiso and Buhirigi, the additional use of relative changes in intensity of infection would have made no difference to classification. Given that Kaiso and Buhirigi are only 30km apart and are within the same district, these differences in classification highlight the need to determine the geographical scale of persistent hotspots to obtain optimal benefits from the finite resources available to national control programmes; a recognised need that has resulted in a change of implementation unit from the district to sub-district level for PC programmes [26].

One caveat is the selection criteria for the schools; chosen to be representative of high infection intensity schools, they were less likely to have reached control programme thresholds and more likely to be considered hotspots for schistosomiasis. It should also be noted that coverage rates in ESPEN were available at the district level and could vary at the village level. Although the Ugandan reporting unit is scheduled to be changed to sub-district, this illustrates a potential logistical difficulty in applying criteria related to treatment coverage to the identification of persistent hotspots. This is particularly true for lake shore communities that are characterised by unstable populations [27]. In Lake Victoria fishing villages in Mayuge District, Uganda, treatment coverage levels as low as 52.6% have been recorded, with age not being a significant predictor of receipt of treatment [28]. That said, Mayuge and Hoima Districts adopt differing targeting strategies, with Mayuge conducting community targeting and Hoima conducting combined school and community targeting of treatment, likely increasing the coverage in the target demographic of in-school SAC most commonly used in PC monitoring.

Another caveat to interpretation is that we used 2016 monitoring data as our "baseline" for assessing whether the communities were persistent hotspots despite a significant number of MDA rounds having been implemented prior to 2016. We were therefore in effect measuring whether annual treatment was having continued significant impact on infection parameters rather than an initial impact. With two thirds of the 51 endemic countries requiring preventative chemotherapy having implemented MDA in all endemic implementation units and only two, Equitorial Guinea and South Africa, not having started MDA by 2020 [4], the reality for most endemic areas will be that hotspot assessment will be conducted in the context of some prior treatment having been provided. Regardless, it is clear that given the history of PC implementation in Hoima District and the apparent failure to attain morbidity control in Buhirigi from both the 2016 monitoring data and the 2020/2021 FibroSchot baseline data, coupled with the evidence that the heavy infection threshold fails to capture the importance of moderate infections in causing poor health [29], there is a continued need for operational research on criteria for identifying "zones of concern" that fail to meet the current definition of persistent hotspot but require alternative or improved control approaches, be they MDA-based or otherwise.

The abundance of fresh water intermediate hosts of schistosomiasis plays a very big role in maintaining the transmission cycle of *S. mansoni* parasites [30]. The higher abundance of *Biomphalaria* species in Kaiso sites compared to those of Buhirigi, is likely attributable to the presence of lentic water flooded lake areas with water hyacinth providing suitable habitat for *Biomphalaria sudanica* and *Biomphalaria pfeifferi* [31,32], while higher abundance of *Biomphalaria stanleyi* at site C in Kaiso community is explainable by presence of submerged *Vallisneria* plants [31]. Only at site A on the River Hoimo were similar habitats observed and here *Biomphalaria pfeifferi* was relatively abundant. Direct transmission from the River Hoimo could be one reason for Buhirigi not reaching the threshold of morbidity control, but also the frequent oscillation of people, including children between the landing sites and Buhirigi to buy fish and visit relatives.

Some studies have shown that adequate sanitation and proper use of latrines lowers *S. mansoni* infections [33,34], although evidence for its impact on schistosomiasis control success is not as strong as that for snail control [4]. Despite this, WASH strategies including improved sanitation are included in the integrated strategies proposed for the elimination of *S. mansoni* by 2030 [1] due to the wider health benefits. Our results indicate inequality in access to good, safe sanitation between the Lake Albert riparian community and the nearby rural agricultural area of Buhirigi. A significant proportion of respondents in Kaiso admitted to open defecation and a significant proportion of those with infants reported burying their faeces rather than safe disposal, pertinent given the known infection of pre-school children in Lake Albert shoreline communities [35,36]. Such poor sanitary conditions are not peculiar to Kaiso but a common feature of Ugandan water bodies landing sites [19,37–39]. The rates of open defecation were lower than reported in these other studies, however,

the questionnaire was applied with answers being exclusive of each other and subtleties of both latrine use and open defecation may not have been recorded. Qualitative analysis of micro and macro-transmission risks in the area does indicate that open defecation is very common in Kaiso [25].

Buhirigi has a variety of water sources while lake water was predominantly used in Kaiso. Infection levels on a micro-geographical scale can be associated with closeness to the permanent water body that acts as the source of domestic water [40–42] and on Lake Albert this is compounded by the interaction between snail host numbers, the snail species present and human host behaviour [43]. The reduced average time taken by the respondents from the Kaiso to collect water implies that the people walk shorter distances and potentially collect water more frequently than the people in Buhirigi community; reflecting an unwillingness by the inhabitants to walk much greater distances to obtain safe water, or a lack of viable safe water options within a walkable distance [25]. Therefore, health education on water collection techniques and the significance of water treatment for home use, including bathing, may help reduce infection.

Finally, the recently published monitoring and evaluation framework for schistosomiasis control programmes incorporates an additional definition for "potential" hotspots [26]. These are communities of ≥10% prevalence where frequent water contact, low coverage of WASH and environmental risk result in potential for normal control measures being insufficient to control morbidity. In Buhirigi, despite the known decrease in infection, given that a) the morbidity control threshold has not been met, b) moderate infections that are associated with clinical morbidity [29] were prevalent and C) that we have established behavioural, WASH and environmental risk factors [25], the outcome of the FibroScHot trial in assessing the impact of increased treatment frequency on morbidity and infection levels in this community will be highly informative as to whether similar "potential" hotspots should be considered "persistent" hotspots and provided with twice per annum treatment when possible.

## Conclusion

We have established that the control programme threshold for morbidity control has not been met in either of our communities despite PC programmes having been administered for more than 15 years. The intermediate snail host collections and the WASH questionnaire results show that there is scope for integrated approaches to be employed towards the attainment of elimination as a public health problem in these communities. Assessing the intensity of infection did not alter the persistence status for either community, indicating that prevalence alone was sufficient to determine hotspot status for these communities. The FibroScHot trial will be able to inform on whether increased frequency of treatment with praziquantel in a *S. mansoni* persistent hotspot and in an area of moderate transmission that has failed to meet morbidity control thresholds will be advantageous in attainment of control programme targets.

## Supporting Information

**S1 Fig. Reported ways of water treatment in Buhirigi and Kaiso communities.** N = 25 (52.1%) in Buhirigi and 26 (52%) in Kaiso answered "yes" to treating their water ($\chi^2 < 0.001$, p = 0.993). Shown are the percentages for how those individuals treated their water. Only answers with >5% of respondents in either community are displayed, with other methods grouped together. Chi-squared analysis of how water was treated: $\chi^2 = 20.527$, p < 0.001.
(PDF)

**S1 Text. STROBE Checklist for cross-sectional studies.**
(DOCX)

## Acknowledgments

The authors would like to thank the FibroScHot participants and their families. They would also like to thank the head teachers and teachers of Kaiso and Buhirigi Primary Schools for all the assistance provided throughout the study. The FibroScHot trial was co-ordinated by the Cambridge Clinical Trials Unit (CCTU) and the authors thank them for the

curation and release of the baseline parasitology data that is used in this manuscript. We also thank Dr H. Curtis Kariuki for providing training in snail identification and cercarial shedding.

## Author contributions

**Conceptualization:** Birgitte J Vennervald, Edridah M. Tukahebwa, Shona Wilson.

**Data curation:** Shona Wilson.

**Formal analysis:** Fred Besigye, Shona Wilson.

**Funding acquisition:** Birgitte J Vennervald, Edridah M. Tukahebwa, Shona Wilson.

**Investigation:** Fred Besigye, Candia Rowel, Moses Adriko.

**Project administration:** Moses Adriko, Birgitte J Vennervald, Fred Nuwaha, Edridah M. Tukahebwa, Shona Wilson.

**Supervision:** Fredrick J Muyodi, John Joseph Kisakye, Rosemary Nalwanga, Shona Wilson.

**Visualization:** Fred Besigye, Shona Wilson.

**Writing – original draft:** Fred Besigye.

**Writing – review & editing:** Candia Rowel, Moses Adriko, Fredrick J Muyodi, John Joseph Kisakye, Rosemary Nalwanga, Birgitte J Vennervald, Fred Nuwaha, Edridah M. Tukahebwa, Shona Wilson.

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
