## [Decision Letter · Decision Letter 0]

9 Mar 2025

PNTD-D-24-01395

School-aged Schistosoma mansoni infection levels after long-term programmatic control show failure to meet control programme targets and evidence of a persistent hotspot: evaluation of the FibroScHot trial baseline data

Dear Dr. Wilson,

Thank you for submitting your manuscript to PLOS Neglected Tropical Diseases. After careful consideration, we feel that it has merit but does not fully meet PLOS Neglected Tropical Diseases's publication criteria as it currently stands. Therefore, we invite you to submit a revised version of the manuscript that addresses the points raised during the review process.

Please submit your revised manuscript within 60 days May 08 2025 11:59PM. If you will need more time than this to complete your revisions, please reply to this message or contact the journal office at plosntds@plos.org. Please include the following items when submitting your revised manuscript:

We look forward to receiving your revised manuscript.

Kind regards,

Francesca Tamarozzi

Section Editor

Shaden Kamhawi

co-Editor-in-Chief

Paul Brindley

co-Editor-in-Chief

**Journal Requirements:**

1) We noticed that you used the phrase 'data not shown' in the manuscript. We do not allow these references, as the PLOS data access policy requires that all data be either published with the manuscript or made available in a publicly accessible database. Please amend the supplementary material to include the referenced data or remove the references.

2) Thank you for including an Ethics Statement for your study. Please state:

i) whether the obtained formal consent is verbal or written

Potential Copyright Issues:

i) Figure 1. Please (a) provide a direct link to the base layer of the map (i.e., the country or region border shape) and ensure this is also included in the figure legend; and (b) provide a link to the terms of use / license information for the base layer image or shapefile. We cannot publish proprietary or copyrighted maps (e.g. Google Maps, Mapquest) and the terms of use for your map base layer must be compatible with our CC BY 4.0 license.

**Reviewers' Comments:**

Reviewer's Responses to Questions

**Key Review Criteria Required for Acceptance?**

**Methods**

-Are the objectives of the study clearly articulated with a clear testable hypothesis stated?

-Is the study design appropriate to address the stated objectives?

-Is the population clearly described and appropriate for the hypothesis being tested?

-Is the sample size sufficient to ensure adequate power to address the hypothesis being tested?

-Were correct statistical analysis used to support conclusions?

-Are there concerns about ethical or regulatory requirements being met?

Reviewer #1: (No Response)

Reviewer #2: A few questions regarding the methods:

1. How were the two 'contrasting' communities (as mentioned in the introduction in line 103-104) selected for this study? You mention how the schools withing these two communities were selected, but it is unclear how the communities itself were selected.

2. In the selection criteria you mention proximity to each other, does this mean schools had to be close together or far apart?

3. To be able to compare to the WHO definition you state that you only used the first stool sample for the analysis, does this mean a single slide from the first stool sample, or two slides from the first stool sample? Earlier in the same paragraph you state that two slides from each sample were prepared ('where possible'). It would be good to be specific on what was used for the analysis (i.e. one slide from a single stool sample).

**Results**

-Does the analysis presented match the analysis plan?

-Are the results clearly and completely presented?

-Are the figures (Tables, Images) of sufficient quality for clarity?

Reviewer #1: (No Response)

Reviewer #2: Major comment regarding Table 1: in the 2016 monitoring exercise, what is the actual the S. mansoni prevalence? The table includes 2x 'negative'?

Please check the numbers in the table carefully: the prevalence numbers do not add up (16+33=49 for Buhirigi, and 34+31=65 for Kaiso?), as well as the intensity of infection numbers for Buhirigi (43+6=49?)

Also, be consistent in the number of decimals used (2 decimals for egg counts not necessary?)

The layout of the table needs improvement.

**Conclusions**

-Are the conclusions supported by the data presented?

-Are the limitations of analysis clearly described?

-Do the authors discuss how these data can be helpful to advance our understanding of the topic under study?

-Is public health relevance addressed?

Reviewer #1: (No Response)

Reviewer #2: The conclusions are supported by the data and the limitations are clearly described. It will be very interesting to see the follow-up data of the trial, whether or not increased frequency of treatment has a better effect on the (intensity of) infection in these affected communities.

**Editorial and Data Presentation Modifications?**

Reviewer #1: (No Response)

Reviewer #2: Minor revision

**Summary and General Comments**

Reviewer #1: It is clear that modified approaches are required for SCH areas that are not responding to treatment. These areas in Uganda are a good case study. As the authors mention they have been receiving treatment for around 15 years and there is still significant infection.

Comments

• When considering the definition of hotspot, my biggest question is around what counts as baseline. In this paper the authors, understandably, use the 2016 monitoring exercise as the de facto baseline. But, as they also point out, there have been more than 15 years of treatment in these areas so the true underlying endemic baseline is likely to be much higher. Is there good data on what the levels of infection were originally, prior to any treatment? If not, are we in danger of using misleading figures?

• Relatedly, if an area has received 10/15 rounds of treatment and not reached control targets, can safely assume it’s an area worthy of greater focus. The levels of infection remain high in both areas so by any sensible classification, both sites require additional interventions. Is there worth in suggesting a new definition for e.g. a ‘zone of concern”? For example, after x years of treatment of at least y% coverage, the threshold has not been met? I’m worried we could get so caught up in whether somewhere meets the definition of hotspot that we miss important areas. And these are undoubtedly important areas.

• It would be worth referencing the work that the trachoma community has done recently ‘persistent’ and ‘recrudescent’ areas and how they have developed modified strategies to reach them, e.g. ‘more frequent than annual’ treatment. There are similarities with the work being done here, and could be some good lessons. ‘Informal consultation on end-game challenges for trachoma elimination’ (https://iris.who.int/bitstream/handle/10665/363591/9789240048089-eng.pdf)

• Conclusion – you say that integrated approaches (including snail control and WASH) will likely be crucial for attainment of elimination. But the paper doesn’t give any data to support that. It’s possible (although I think the evidence base is equivocal) but not explored here.

• Check numbers and percentages in Table 1 – especially for top S.mansoni prevalence rows. Te numbers do not add up for Kaiso (34+31 don’t equal 51). ‘Negative’ has been repeated.

• Prevalence categories – would it make more sense to have three categories: Uninfected, light/moderate, heavy? Or even four: Uninfected, light, moderate, heavy. Rather than collapsing uninfected with light and moderate. A real difference between 0 and 399 epg!

• Sample size – how useful is the 2016 baseline data if there are only 50 children in two schools? Is that sufficient for study? Relatedly, how useful is the Bhugiri baseline parasitology data if there was a year of treatment between them?

• Increase in infections in Kaiso is striking – both prevalence and intensity – what do you think the reasons are for this?

Reviewer #2: I read the manuscript with great interest. The FibroScHot trial is highly relevant, and in this first paper the authors describe the baseline findings and discuss whether the study sites should be considered as a hotspot based on WHO definitions and how this can influence the analysis of the trial results.

PLOS authors have the option to publish the peer review history of their article (what does this mean? ). If published, this will include your full peer review and any attached files.

**Do you want your identity to be public for this peer review?** For information about this choice, including consent withdrawal, please see our Privacy Policy .

Reviewer #1: **Yes: ** Michael French

Reviewer #2: **Yes: ** Pytsje Hoekstra

**Figure resubmission:**
---

## [Decision Letter · Decision Letter 1]

3 May 2025

Dear Dr Wilson,

We are pleased to inform you that your manuscript 'School-aged Schistosoma mansoni infection levels after long-term programmatic control show failure to meet control programme targets and evidence of a persistent hotspot: evaluation of the FibroScHot trial baseline data' has been provisionally accepted for publication in PLOS Neglected Tropical Diseases.

Best regards,

Francesca Tamarozzi

Section Editor

Francesca Tamarozzi

Section Editor

Shaden Kamhawi

co-Editor-in-Chief

Paul Brindley

co-Editor-in-Chief

Reviewer's Responses to Questions

**Key Review Criteria Required for Acceptance?**

**Methods**

-Are the objectives of the study clearly articulated with a clear testable hypothesis stated?

-Is the study design appropriate to address the stated objectives?

-Is the population clearly described and appropriate for the hypothesis being tested?

-Is the sample size sufficient to ensure adequate power to address the hypothesis being tested?

-Were correct statistical analysis used to support conclusions?

-Are there concerns about ethical or regulatory requirements being met?

Reviewer #1: (No Response)

Reviewer #2: (No Response)

**Results**

-Does the analysis presented match the analysis plan?

-Are the results clearly and completely presented?

-Are the figures (Tables, Images) of sufficient quality for clarity?

Reviewer #1: (No Response)

Reviewer #2: (No Response)

**Conclusions**

-Are the conclusions supported by the data presented?

-Are the limitations of analysis clearly described?

-Do the authors discuss how these data can be helpful to advance our understanding of the topic under study?

-Is public health relevance addressed?

Reviewer #1: (No Response)

Reviewer #2: (No Response)

**Editorial and Data Presentation Modifications?**

Reviewer #1: (No Response)

Reviewer #2: (No Response)

**Summary and General Comments**

Reviewer #1: Thank you for the responses to my initial comments. Good luck with publication.

Reviewer #2: The manuscript has improved greatly with the adjustments the authors have made. I therefore recommend to accept the manuscript for publication.

PLOS authors have the option to publish the peer review history of their article (what does this mean? ). If published, this will include your full peer review and any attached files.

**Do you want your identity to be public for this peer review?** For information about this choice, including consent withdrawal, please see our Privacy Policy .

Reviewer #1: **Yes: ** Michael French

Reviewer #2: **Yes: ** PT Hoekstra

---

## [Editor Report · Acceptance letter]

Dear Dr Wilson,

We are delighted to inform you that your manuscript, "School-aged Schistosoma mansoni infection levels after long-term programmatic control show failure to meet control programme targets and evidence of a persistent hotspot: evaluation of the FibroScHot trial baseline data," has been formally accepted for publication in PLOS Neglected Tropical Diseases.

Best regards,

Shaden Kamhawi

co-Editor-in-Chief

Paul Brindley

co-Editor-in-Chief
